# LIFT: Likelihood-Based Intervention Faithfulness Training for Chain-of-Thought in Language Models

## Abstract

Chain-of-thought (CoT) reasoning improves LLM accuracy on complex tasks, but CoTs are often unfaithful; that is, the model does not rely on the stated reasoning steps to produce its answer. Existing methods to improve faithfulness require human annotations or are limited to small models. We propose LIFT, which automatically scores each reasoning step by its causal influence: we replace it with a domain-matched but irrelevant fact and measure the drop in answer log-probability. We fine-tune on high-scoring CoT traces where interventions successfully disrupted reasoning, jointly optimizing for faithfulness and accuracy. LIFT improves faithfulness across three models (Mistral-7B-Instruct, Qwen3-8B, and Qwen3-4B-Instruct) and three benchmarks, with no significant changes in accuracy.

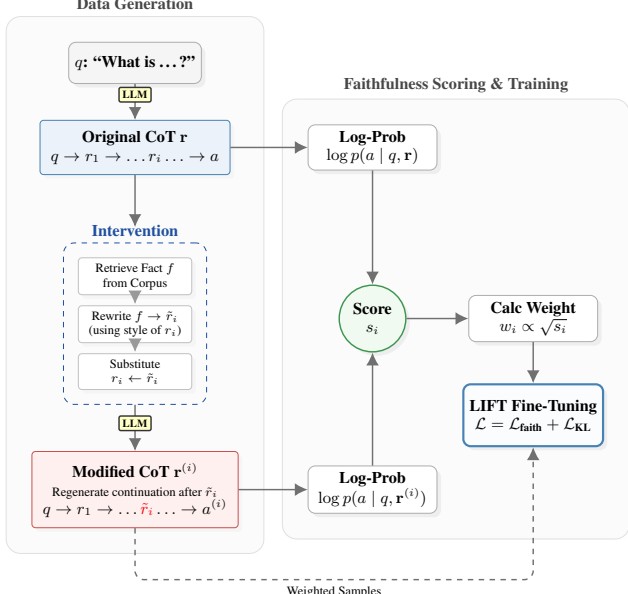

*Figure 1.* LIFT training pipeline for causal chain-of-thought faithfulness.

## 1. Introduction

Chain-of-thought (CoT) reasoning has become a standard technique for solving complex problems with large language models (LLMs). By generating step-by-step reasoning before producing a final answer, CoTs improve accuracy on multi-step tasks (Wei et al., 2022) and are widely interpreted as explanations of the model's decision process. This interpretability is valuable for debugging model failures, auditing reasoning in high-stakes applications, and building user trust.

However, CoTs are frequently *unfaithful*: recent work shows the final answer often does not causally depend on the intermediate reasoning steps (Lanham et al., 2023; Barez et al.). When reasoning is unfaithful, users may trust flawed explanations, developers cannot identify when models use shortcuts or biases, and oversight becomes difficult in safety-critical deployments. The challenge is to train models that generate CoTs which are both accurate *and* causally linked to their answers, without requiring expensive human annotation of faithful and unfaithful reasoning.

Existing approaches have significant limitations. Methods like FRODO (Paul et al., 2024) use LLM-generated counterfactual data and Direct Preference Optimization to tailor small-sized language models (under 10B parameters) to generate correct reasoning steps and robustly reason over them, but do not provide a scalable approach for arbitrary LLM sizes. Additionally, FRODO relies on manual human filtering of generated traces, which does not scale easily for large quantities of data. Other methods similar to FRODO fail to achieve significant gains in faithfulness (Swaroop et al., 2025). While some, like Lightman et al. (2023), have used extensive human annotations, releasing PRM800K with 800,000 step-level human feedback labels for process supervision, such methods require costly manual labeling. Most prior work focuses on *measuring* faithfulness (Lanham et al., 2023) rather than improving it through training.

[1]Anonymous Institution, Anonymous City, Anonymous Region, Anonymous Country. Correspondence to: Anonymous Author <anon.email@domain.com>.

Preliminary work. Under review by the International Conference on Machine Learning (ICML). Do not distribute.

No existing method provides a scalable, automated way to fine-tune arbitrary LLMs for improved CoT faithfulness.

We introduce LIFT (**L**ikelihood-based **I**ntervention **F**aithfulness **T**raining), a fully automated method to improve CoT faithfulness through counterfactual interventions. Our key insight is that faithful reasoning steps should be *causally necessary*: replacing a step with an irrelevant fact should change the model's answer. We formalize this by measuring log-probability drops after interventions. For each step, we replace it with a domain-matched but irrelevant fact from a fact corpus, regenerate the CoT continuation, and measure how much the original answer's log-probability decreases. Larger drops indicate the step was causally important. We use these scores to select training data (keeping examples where interventions disrupted reasoning) and weight them during fine-tuning (prioritizing high-scoring examples). To maintain accuracy, we jointly optimize on faithfulness-scored traces and final-answer accuracy, with a hyperparameter $\eta$ controlling the trade-off.

We fine-tune three models with LIFT (Mistral-7B-Instruct-v0.3, Qwen3-8B, and Qwen3-4B-Instruct-2507) and evaluate across three reasoning benchmarks: GSM8K, SVAMP, and StrategyQA. LIFT produces substantial and consistent improvements in faithfulness across all models and tasks while preserving answer accuracy. For example, Mistral-7B nearly doubles its faithfulness on GSM8K, and Qwen3-4B more than triples its faithfulness on StrategyQA, yet both models maintain accuracy comparable to their base versions. These results demonstrate that models can be trained to generate reasoning traces where intermediate steps genuinely influence the final answer, without sacrificing their ability to solve problems correctly.

**Contributions.**

1. We introduce a counterfactual intervention method that automatically scores CoT steps for faithfulness by measuring their causal influence on the model's answer, requiring no human annotation.

2. We present LIFT, a fine-tuning procedure using these scores to improve faithfulness while maintaining accuracy, with a tunable parameter controlling the trade-off.

3. We demonstrate faithfulness improvements across three models (Mistral-7B-Instruct-v0.3, Qwen3-8B, Qwen3-4B-Instruct-2507) and three benchmarks (GSM8K, SVAMP, StrategyQA), achieving relative faithfulness increases of 20–250% while preserving accuracy.

4. We perform ablations on answer weight during fine-tuning to demonstrate a trade-off between accuracy and faithfulness.

## 2. Background

We consider a language model $\mathcal{M}$ that generates chain-of-thought (CoT) reasoning traces to solve problems. Given a question $q$, the model generates a reasoning trace $\mathbf{r} = (r_1, \ldots, r_n)$ of $n$ steps, followed by a final answer $a$.

**Faithfulness.** A reasoning trace is *faithful* if the model's answer causally depends on the intermediate reasoning steps. We measure faithfulness by performing counterfactual interventions: modifying step $r_i$ to produce a modified trace $\mathbf{r}^{(i)}$, then measuring the log-probability drop:

$$\text{Faithfulness}(r_i) = \log p(a \mid q, \mathbf{r}) - \log p(a \mid q, \mathbf{r}^{(i)}) \quad (1)$$

Higher scores indicate greater causal influence.

**Fact Corpus.** In order to retrieve facts for interventions, we construct a corpus $\mathcal{F}$ of arithmetic equations and Wikidata facts, clustered into $K$ semantic domains using k-means on embeddings (details in Appendix A).

## 3. Method

LIFT consists of interventions to replace steps with counterfactual statements, scoring to measure faithfulness, and fine-tuning to increase the faithfulness of the model.

### 3.1. Intervention

Given trace $\mathbf{r} = (r_1, \ldots, r_n)$ and step $i$, we replace $r_i$ with a domain-matched but irrelevant fact via two-stage retrieval: first, select the cluster $k^*$ most similar to $r_i$, then select the fact $f^* \in C_{k^*}$ least similar to $r_i$.

We rewrite $f^*$ to match the writing style of $r_i$ using the model (in order to maintain a natural flow of text), producing $\tilde{r}_i$ (details in Appendix B).

$\text{emb}(\cdot)$ denotes embeddings, $S_C(A, B)$ denotes cosine similarity ($S_C(A, B) = \frac{A \cdot B}{|A||B|}$), and $\mu_k$ denotes the centroid of embeddings in cluster $k$.

---

**Algorithm 1** Intervene

---

**Require:** Trace $\mathbf{r}$, index $i$, corpus $\mathcal{F}$ with clusters $\{C_k\}$
**Ensure:** Intervened step $\tilde{r}_i$
1: $k^* = \operatorname{argmax}_{k \in \{1, 2, \ldots\}} S_C(\text{emb}(r_i), \mu_k)$
2: $f^* = \operatorname{argmin}_{f \in C_{k^*}} S_C(\text{emb}(r_i), \text{emb}(f))$
3: $\tilde{r}_i = \text{StyleRewrite}(f^*, r_i)$
4: **return** $\tilde{r}_i$

---

### 3.2. Scoring

We measure faithfulness by intervention and regeneration. After replacing $r_i$ with $\tilde{r}_i$, we regenerate the continuation

$$(\hat{r}_{i+1}, \ldots, \hat{r}_m, a^{(i)}) \sim \mathcal{M}(\cdot \mid q, r_1, \ldots, r_{i-1}, \tilde{r}_i), \quad (2)$$

by sampling (temperatures are listed in Section 4), forming $\mathbf{r}^{(i)} = (r_1, r_2, \ldots, r_{i-1}, \tilde{r}_i, \hat{r}_{i+1}, \ldots, \hat{r}_m)$, where $m$ is the number of steps in the regenerated trace (which may differ from $n$, the original number of steps). The score is

$$s_i = \log p(a \mid q, \mathbf{r}) - \log p(a \mid q, \mathbf{r}^{(i)}) \qquad (3)$$

Positive scores indicate successful disruption; a score of $s_i$ means that $\mathbf{r}^{(i)}$ is $e^{s_i}$ times less likely than the original trace $\mathbf{r}$ to generate the original final answer $a$.

---

**Algorithm 2** Score

---

**Require:** Question $q$, trace $\mathbf{r} = (r_1, \ldots, r_n)$, answer $a$, model $\mathcal{M}$
**Ensure:** Scores $(s_1, \ldots, s_n)$, modified traces $\{(\mathbf{r}^{(i)}, a^{(i)})\}$
1: $\ell_{\text{base}} = \log p_{\mathcal{M}}(a \mid q, \mathbf{r})$
2: **for** $i = 1$ to $n$ **do**
3: $\quad \tilde{r}_i = \text{Intervene}(\mathbf{r}, i, \mathcal{F})$
4: $\quad (\hat{r}_{i+1}, \ldots, \hat{r}_m, a^{(i)}) \sim \mathcal{M}(\cdot \mid q, r_1, \ldots, r_{i-1}, \tilde{r}_i)$
5: $\quad \mathbf{r}^{(i)} = (r_1, r_2, \ldots, r_{i-1}, \tilde{r}_i, \hat{r}_{i+1}, \ldots, \hat{r}_m)$
6: $\quad \ell_{\text{mod}}^{(i)} = \log p_{\mathcal{M}}(a \mid q, \mathbf{r}^{(i)})$
7: $\quad s_i = \ell_{\text{base}} - \ell_{\text{mod}}^{(i)}$
8: **end for**
9: **return** $(s_1, \ldots, s_n)$, $\{(\mathbf{r}^{(i)}, a^{(i)})\}$

---

**Input** ($q$): A farmer has 15 apples. He gives 5 to his neighbor. He sells three at the market. How many does he have left?

**Original Trace** ($\mathbf{r}$)     $\ln(p(a|\mathbf{r})) = -0.04$
$r_1$: The farmer starts with 15 apples.
$r_2$: He gives away 5 apples, so 15 - 5 = 10.
$r_3$: He sells 3 apples, so 10 - 3 = 7.
$a$: 7

**Intervention**     **Faithfulness Score** $s_2$
$\ell_{base} - \ell_{mod} = 3.87$

**Intervened Trace** ($\mathbf{r}^{(2)}$)     $\ln(p(a|\mathbf{r}^{(2)})) = -3.91$
$r_1$: The farmer starts with 15 apples.
$\tilde{r}_2$: *He gives away 6 apples, so 15 - 6 = 9*
$\hat{r}_3$: [Regenerated: He sells 3 apples, so 9 - 3 = 6.]
$a^{(2)}$: 6

*Figure 2.* An example chain-of-thought intervention and score.

### 3.3. Fine-Tuning

**Data Generation.** For each question $q_j$ with ground truth $a_{\text{true},j}$, we sample a trace $(\mathbf{r}_j, a_j) \sim \mathcal{M}(\cdot \mid q_j)$ (temperatures listed in Section 4), then intervene on each step, generating modified traces $\{(\mathbf{r}_j^{(i)}, a_j^{(i)})\}$ and scores (with $s_j^{(i)}$ corresponding to the trace with step $i$ of $\mathbf{r}_j$ modified). We

filter traces to be in the range $[s_{\text{low}}, s_{\text{high}}]$ (low-scoring traces likely provide too weak a signal to have a significant impact on faithfulness, and extremely high-scoring traces are likely gibberish that decreases the log probability by being nonsensical), normalize to the range $[0.05, 1]$ to avoid minuscule weights, and take the square root to prevent especially high-scoring traces from dominating.

$$w_j^{(i)} = \sqrt{0.05 + 0.95 \cdot \frac{s_i^{(j)} - s_{\text{min}}}{s_{\text{max}} - s_{\text{min}}}} \qquad (4)$$

For $w_j^{(i)}$ greater than a certain threshold $\tau$, we append ground truth answers to the trace for accuracy tuning.

Token $t$ receives weight

$$\omega_j^{(i)}(t) = \begin{cases} 0 & \text{if input (prompt) token,} \\ 0 & \text{if reasoning token at or before } \tilde{r}_j^{(i)}, \\ w_j^{(i)} & \text{if reasoning token after } \tilde{r}_j^{(i)}, \\ \eta \cdot w_j^{(i)} & \text{if answer token,} \end{cases}$$

$$(5)$$

where $\eta$ is a hyperparameter controlling how strongly to weight accuracy.

Assigning weights in this manner ensures the model isn't trained to generate irrelevant facts, but rather to generate faithful reasoning *after* generating an incorrect step.

**Objective.** We optimize $\mathcal{L} = \mathcal{L}_{\text{faithfulness}} + \mathcal{L}_{\text{KL}}$, where

$$\mathcal{L}_{\text{faithfulness}} = \frac{1}{|\mathcal{D}|} \sum_{(j,i) \in \mathcal{D}} \frac{\sum_t \omega_j^{(i)}(t) \ell_t}{\sum_t \omega_j^{(i)}(t)} \qquad (6)$$

$$\mathcal{L}_{\text{KL}} = \frac{\beta}{|\mathcal{D}|} \sum_{(j,i) \in \mathcal{D}} \frac{\sum_t \omega_j^{(i)}(t) D_{\text{KL}}(p_{\mathcal{M}_{\text{ref}}} \| p_{\mathcal{M}})}{\sum_t \omega_j^{(i)}(t)} \qquad (7)$$

with $\ell_t$ the cross-entropy at token $t$, $\mathcal{M}_{\text{ref}}$ the frozen base model (that is, the model before fine-tuning), and $\beta$ controlling regularization strength.

We fine-tune for one epoch using a LoRA adapter (necessary in order to fit on VRAM). Hyperparameters (clustering $K$, LoRA rank, decoding temperatures, and hyperparameters for fine-tuning) were chosen by hand-tuning to produce stable training behavior on a small development set. The exact values used in all experiments are listed in Section 4.

## 4. Experimental Setup

We evaluate LIFT across multiple model architectures and reasoning benchmarks to assess both faithfulness improvements and accuracy preservation. This section describes our model selection, computational infrastructure, data generation procedure, training configuration, evaluation methodology, and ablation study design.

---

**Algorithm 3** LIFT Fine-Tuning

---

**Require:** Questions $\{q_j\}$, ground-truth answers $\{a_{\text{true},j}\}$, model $\mathcal{M}$, weight $\beta$, score range $[s_{\text{low}}, s_{\text{high}}]$, answer score threshold $\tau$

**Ensure:** Fine-tuned model $\mathcal{M}'$

1: $\mathcal{D} = \emptyset$
2: **for** each $q_j$ **do**
3:     $(\mathbf{r}_j, a_j) \sim \mathcal{M}(\cdot \mid q_j)$
4:     $\{(\mathbf{r}_j^{(i)}, a_j^{(i)}, s_j^{(i)})\} = \text{Score}(q_j, \mathbf{r}_j, a_j)$
5:     **for** $i$ with $s_i^{(j)} \in [s_{\text{low}}, s_{\text{high}}]$ **do**
6:         Compute $w_j^{(i)}$ via Equation (4)
7:         **if** $w_j^{(i)} \geq \tau$ **then**
8:             Append "Answer: $a_{\text{true},j}$" to $\mathbf{r}_j^{(i)}$
9:         **end if**
10:        Compute token-level weights $\omega_j^{(i)}$ using Equation (5)
11:        Append $(q_j, \mathbf{r}_j^{(i)}, \omega_j^{(i)})$ to $\mathcal{D}$
12:    **end for**
13: **end for**
14: Fine-tune $\mathcal{M}$ on $\mathcal{D}$ using Equations (6) and (7)
15: **return** $\mathcal{M}'$

---

**Models.** We apply LIFT to three pretrained language models of varying sizes and architectures: Mistral-7B-Instruct-v0.3, Qwen3-8B, and Qwen3-4B-Instruct-2507. We apply LIFT using LoRA adapters (Hu et al.) with rank $r = 64$, LoRA alpha $\alpha = 128$, and dropout 0.05, targeting all linear projection layers in the attention and feedforward blocks. This adapter-based approach enables efficient fine-tuning while preserving the base model's general capabilities.

**Training Data Generation.** We construct training datasets by sampling from the train splits of six reasoning benchmarks: GSM8K (Cobbe et al., 2021), SVAMP (Patel et al., 2021), StrategyQA (Geva et al., 2021), CommonsenseQA (Talmor et al., 2019), ASDiv (Miao et al., 2020), and SciBench (Wang et al., 2024). For each model, we randomly select 1000 prompts from the combined pool of all six datasets. For each selected question, we generate a preliminary reasoning trace using the model. Temperature settings during this phase vary by model: we use temperature $T = 0.1$ for Mistral-7B-Instruct-v0.3 and $T = 0.5$ for both Qwen models. These settings balance the need for quality initial traces with sufficient diversity across examples.

After generating preliminary traces, we apply Algorithm 2 to produce intervened traces and faithfulness scores for each reasoning step. When generating continuations after intervened steps, we use higher temperatures to encourage diverse reasoning paths: temperature $T = 0.2$ for the Mistral model and temperature $T = 1.6$ for both Qwen models. The elevated temperature for continuation generation increases

the likelihood that the model will explore alternative reasoning directions after the intervention rather than attempting to recover the original answer despite the disrupted step.

We set $[s_{\text{low}}, s_{\text{high}}] = [0.5, 12]$, as traces with faithfulness scores below 0.5 typically ignore the intervened step or self-correct, and those with scores above 12 are typically incoherent.

**Training Configuration.** We train using the AdamW optimizer with model-specific learning rates: $7 \cdot 10^{-5}$ for Mistral-7B and $3 \cdot 10^{-4}$ for both Qwen models. We use a per-device batch size of 4 for the Mistral model and 2 for both Qwen models with gradient accumulation over 2 steps. All models are trained for one epoch using mixed precision (fp16) to reduce memory consumption and accelerate training. The KL regularization coefficient $\beta$ in Equation 7 is set to 0.8 for Mistral-7B and 2.0 for Qwen models.

We use different values of $\eta$ depending on the specific model and training dataset (see Table 1). We use one value for GSM8K, one for StrategyQA and CommonsenseQA, and one for the remaining three datasets.

| Model | GSM8K | QA Tasks | Other |
|---|---|---|---|
| Mistral-7B[3] | 1.8 | 0.4 | 0.8 |
| Qwen3-8B | 1.0 | 2.5 | 3.0 |
| Qwen3-4B[3] | 1.0 | 2.5 | 3.0 |

*Table 1.* $\eta$ hyperparameter by model and dataset type.

**Evaluation Benchmarks and Protocol.** We evaluate models on the test splits of GSM8K, SVAMP, and StrategyQA. These benchmarks span different reasoning types and difficulty levels. GSM8K and SVAMP require multi-step arithmetic with numerical answers, while StrategyQA requires fact composition with boolean answers. We use the first 480 entries from the test splits of GSM8K and StrategyQA, and the entire test split of SVAMP, which contains 300 entries.

For each model, we evaluate three configurations: the base model prompted with chain-of-thought reasoning, the base model prompted without chain-of-thought (direct answer prediction), and the LIFT-aligned model prompted with chain-of-thought reasoning. All evaluations use greedy decoding to ensure deterministic outputs. We assess answer accuracy by comparing the model's final answer to the ground truth using exact-match comparison. For GSM8K and SVAMP, we extract the final numeric value from the model's output. For StrategyQA, we extract the boolean answer (True or False) via case-insensitive pattern matching.

**Faithfulness Measurement.** For all chain-of-thought evaluations, we measure faithfulness using our intervention-based metric from Algorithm 2. For each reasoning step $r_i$ in a generated trace $\mathbf{r} = (r_1, \ldots, r_n)$, we apply Algorithm 1 to produce an intervened step $\tilde{r}_i$. For each intervened step,

*Table 2.* Performance comparison across all tested models. 95.4% confidence intervals are reported ($\pm 2SE$).

| Model | Method | Metric | GSM8K | SVAMP | StrategyQA |
|---|---|---|---|---|---|
| Mistral-7B[3] | Aligned (CoT) | Accuracy | $0.41 \pm 0.05$ | $\mathbf{0.69 \pm 0.04}$ | $\mathbf{0.84 \pm 0.03}$ |
| | | Faithfulness | $\mathbf{0.51 \pm 0.02}$ | $\mathbf{0.42 \pm 0.02}$ | $\mathbf{0.33 \pm 0.02}$ |
| | | Static Faithfulness | $\mathbf{0.43 \pm 0.02}$ | $\mathbf{0.42 \pm 0.02}$ | $\mathbf{0.22 \pm 0.02}$ |
| | Base (CoT) | Accuracy | $\mathbf{0.47 \pm 0.05}$ | $0.61 \pm 0.05$ | $0.83 \pm 0.04$ |
| | | Faithfulness | $0.26 \pm 0.02$ | $0.35 \pm 0.02$ | $0.28 \pm 0.02$ |
| | | Static Faithfulness | $0.25 \pm 0.02$ | $0.30 \pm 0.02$ | $0.17 \pm 0.02$ |
| | Raw | Accuracy | $0.08 \pm 0.03$ | $0.49 \pm 0.05$ | $0.72 \pm 0.05$ |
| Qwen3-8B | Aligned (CoT) | Accuracy | $\mathbf{0.86 \pm 0.03}$ | $\mathbf{0.95 \pm 0.02}$ | $\mathbf{0.91 \pm 0.03}$ |
| | | Faithfulness | $\mathbf{0.21 \pm 0.01}$ | $\mathbf{0.23 \pm 0.01}$ | $\mathbf{0.20 \pm 0.01}$ |
| | | Static Faithfulness | $\mathbf{0.16 \pm 0.01}$ | $\mathbf{0.18 \pm 0.01}$ | $\mathbf{0.14 \pm 0.01}$ |
| | Base (CoT) | Accuracy | $0.84 \pm 0.04$ | $0.94 \pm 0.02$ | $0.90 \pm 0.03$ |
| | | Faithfulness | $0.17 \pm 0.01$ | $0.16 \pm 0.01$ | $0.14 \pm 0.02$ |
| | | Static Faithfulness | $0.13 \pm 0.01$ | $0.16 \pm 0.01$ | $0.09 \pm 0.01$ |
| | Raw | Accuracy | $0.17 \pm 0.04$ | $0.68 \pm 0.05$ | $0.40 \pm 0.05$ |
| Qwen3-4B[3] | Aligned (CoT) | Accuracy | $0.90 \pm 0.03$ | $\mathbf{0.93 \pm 0.02}$ | $0.86 \pm 0.03$ |
| | | Faithfulness | $\mathbf{0.18 \pm 0.01}$ | $\mathbf{0.16 \pm 0.01}$ | $\mathbf{0.14 \pm 0.01}$ |
| | | Static Faithfulness | $\mathbf{0.14 \pm 0.01}$ | $\mathbf{0.16 \pm 0.01}$ | $\mathbf{0.12 \pm 0.01}$ |
| | Base (CoT) | Accuracy | $\mathbf{0.93 \pm 0.02}$ | $0.91 \pm 0.03$ | $\mathbf{0.87 \pm 0.03}$ |
| | | Faithfulness | $0.13 \pm 0.01$ | $0.10 \pm 0.01$ | $0.04 \pm 0.01$ |
| | | Static Faithfulness | $0.08 \pm 0.01$ | $0.11 \pm 0.01$ | $0.04 \pm 0.01$ |
| | Raw | Accuracy | $0.33 \pm 0.04$ | $0.70 \pm 0.04$ | $0.57 \pm 0.05$ |

we generate a continuation and compute the faithfulness score $s_i$ as specified in Equation 1. The overall faithfulness score for a trace is computed as the mean of stepwise scores, $\frac{1}{n} \sum_{i=1}^{n} s_i$.

In addition to our primary faithfulness metric, we evaluate static faithfulness. Static faithfulness measures the log-probability drop when replacing step $r_i$ with $\tilde{r}_i$ but keeping all subsequent steps $r_{i+1}, \ldots, r_n$ unchanged from the original trace.

Static faithfulness provides a more conservative estimate of causal influence, measuring whether a step's specific content directly affects the answer probability even when forcing the model to continue with the original reasoning path. This contrasts with our primary metric, which allows the model to naturally diverge after intervention. We provide additional discussion of static faithfulness and its interpretation in Appendix C.

We provide details on computational cost and approximate times in Appendix D.

## 5. Results

Our evaluation results are summarized in Table 2. Figure 3 displays accuracy and faithfulness results[1]. The LIFT-aligned models show increases in faithfulness compared to the base model across all base models and datasets, statistically significant[2] in all cases except Mistral-7B on StrategyQA.

These results demonstrate a marked improvement in faithfulness, with relative increases in faithfulness ranging from 20% to 250% depending on the model and benchmark. Mistral-7B exhibits large gains, nearly doubling its faithfulness score on GSM8K (96% increase from 0.26 to 0.51) and achieving a 20% improvement on SVAMP. Qwen3-4B demonstrates remarkable growth on StrategyQA with

---

[1]Note that, although all dataset-average faithfulness values in our experiments were in the range $[0, 1]$, they are not fractions and therefore may be greater than 1.

[2]Statistical significance determined by non-overlapping 95.4% confidence intervals.

[3]Model names are shortened to prevent overflow. "Mistral-7B" refers to Mistral-7B-Instruct-v0.3, and "Qwen3-4B" refers to Qwen3-4B-Instruct-2507.

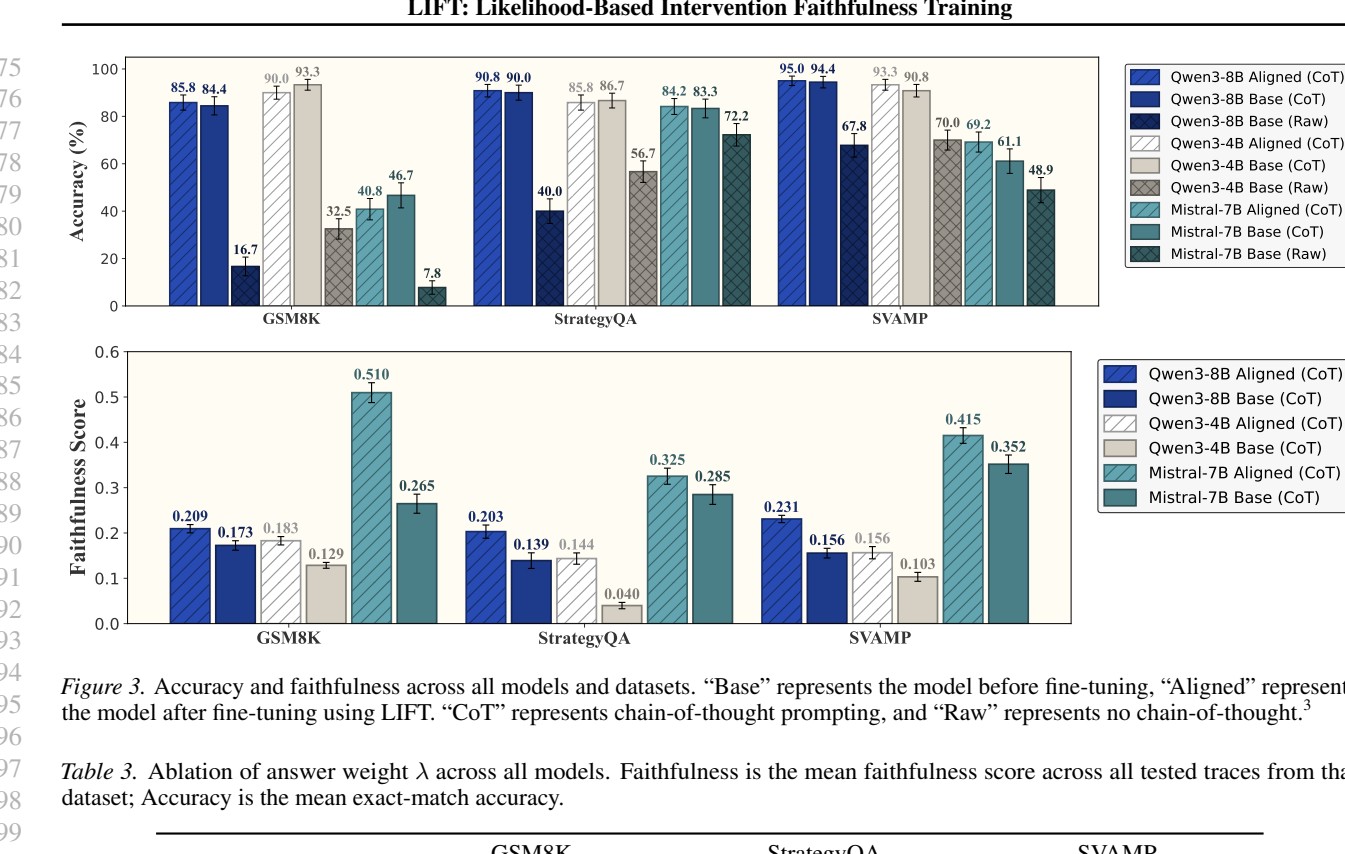

*Figure 3.* Accuracy and faithfulness across all models and datasets. "Base" represents the model before fine-tuning, "Aligned" represents the model after fine-tuning using LIFT. "CoT" represents chain-of-thought prompting, and "Raw" represents no chain-of-thought.[3]

*Table 3.* Ablation of answer weight $\lambda$ across all models. Faithfulness is the mean faithfulness score across all tested traces from that dataset; Accuracy is the mean exact-match accuracy.

| Model | $\lambda$ | GSM8K | | StrategyQA | | SVAMP | |
|---|---|---|---|---|---|---|---|
| | | Faithfulness | Accuracy | Faithfulness | Accuracy | Faithfulness | Accuracy |
| Mistral-7B[3] | 0.0 | 0.685 | 0.241 | 0.373 | 0.667 | 0.558 | 0.600 |
| | 0.5 | 0.556 | 0.433 | 0.388 | 0.766 | 0.461 | 0.600 |
| | 1.0 | 0.510 | 0.408 | 0.325 | 0.842 | 0.415 | 0.692 |
| Qwen3-8B | 0.0 | 0.450 | 0.400 | 0.423 | 0.600 | 0.470 | 0.633 |
| | 0.5 | 0.260 | 0.833 | 0.185 | 0.899 | 0.240 | 0.934 |
| | 1.0 | 0.191 | 0.883 | 0.186 | 0.908 | 0.228 | 0.925 |
| Qwen3-4B[3] | 0.0 | 0.229 | 0.767 | 0.163 | 0.834 | 0.214 | 0.867 |
| | 0.5 | 0.221 | 0.866 | 0.133 | 0.900 | 0.176 | 0.866 |
| | 1.0 | 0.183 | 0.900 | 0.144 | 0.858 | 0.157 | 0.933 |

a 250% relative increase (from 0.04 to 0.14), alongside a 60% improvement on SVAMP. Qwen3-8B shows more modest but consistent improvements, with a 44% increase on SVAMP and 43% on StrategyQA.

Critically, these faithfulness improvements are achieved without compromising task performance. Across all three models, we find no significant change in accuracy between the base model with CoT and the aligned model with CoT.

The results in Table 2 show a consistent increase in static faithfulness across all evaluated models and benchmarks. In Mistral-7B, static faithfulness improved from 0.25 to 0.43 on GSM8K and from 0.30 to 0.42 on SVAMP. Similarly, Qwen3-4B demonstrated a gain from 0.04 to 0.12 on StrategyQA. As defined in Appendix C, static faithfulness measures the log-probability drop of the answer when a step

is replaced with a counterfactual edit while keeping the subsequent steps fixed. An increase in this metric shows that after LIFT fine-tuning, individual steps have a greater impact on the final answer, even without affecting subsequent steps.

These improvements are noteworthy because LIFT does not explicitly train for static faithfulness. As described in Algorithm 3, the fine-tuning data $\mathcal{D}$ consists of modified traces where the model generates a new continuation following an intervention. Consequently, the model is trained to maintain logical consistency during active regeneration rather than under the fixed-suffix conditions used in static evaluation. The observed gains in static faithfulness suggest that the faithfulness gains through LIFT lead to generalization beyond one specific faithfulness measure.

## 5.1. Ablation Study

To characterize the trade-off between faithfulness and accuracy in LIFT, we perform an ablation study over a parameter $\lambda$, which scales the coefficient $\eta$ in Equation 5. We train each model with three configurations: $\lambda = 0.0$ (pure faithfulness training with no answer supervision), $\lambda = 1.0$ (balanced training using our default answer weight settings; this is the same as the aligned models shown in Table 2), and $\lambda = 0.5$, weighing answers half as strongly as the balanced training. The $\lambda = 0.0$ configuration shows the level of faithfulness that can be achieved using LIFT if accuracy degradation is not a concern, while $\lambda = 1.0$ is our recommended default that balances faithfulness improvements with answer accuracy preservation.

For each ablation configuration, we follow the same data generation and training procedures described in Section 3, changing only the hyperparameter $\eta$ to $\lambda \cdot \eta$. We evaluate all configurations on the same test sets using identical faithfulness and accuracy metrics.

The ablation results are shown in Table 3. They provide empirical evidence for a controllable trade-off between the faithfulness of reasoning steps and the model's final answer accuracy. At $\lambda = 0.0$, where the model is fine-tuned solely on faithfulness, we observe the highest faithfulness scores across all datasets. For example, Mistral-7B achieves a faithfulness score of 0.685 on GSM8K at $\lambda = 0.0$, compared to 0.510 at $\lambda = 1.0$. However, this focus on faithfulness significantly reduces accuracy, particularly on arithmetic tasks like GSM8K, where Mistral's accuracy drops to 0.241.

As $\lambda$ increases, the inclusion of ground-truth answer supervision allows the models to recover significant task performance. In the case of Qwen3-8B on GSM8K, moving from $\lambda = 0.0$ to $\lambda = 0.5$ results in a 43.3-point increase in accuracy (0.400 to 0.833), albeit with a drop in faithfulness, from 0.450 to 0.260. This suggests that a greater $\lambda$ results in greater accuracy at the cost of lower faithfulness.

This ablation demonstrates that LIFT provides a mechanism to adjust the faithfulness-accuracy trade-off based on application requirements. Applications requiring maximum interpretability and causal transparency (such as safety-critical decision support systems) may prefer higher faithfulness at some accuracy cost, while applications prioritizing task performance may prefer configurations that maintain stronger answer supervision.

## 5.2. Qualitative Example

In order to illustrate the differences in reasoning style between models aligned with LIFT and base models, we provide an example of reasoning traces from Qwen3-4B-Instruct-2507 responding to a chain-of-thought prompt. Faithfulness scores per step are shown.

---

**Question:** Happy hardcore is a music genre of hard dance. Happy hardcore emerged both from the UK breakbeat hardcore rave scene, and Belgian, German and Dutch hardcore techno scenes. A funeral is traditionally a somber event. Funerals typically do not involve dancing. Raves are typically energetic and upbeat places and are not somber like a funeral. Would it be unusual to play Happy hardcore music at a funeral?

**Base Model (Qwen3-4B-Instruct-2507)**

*Reasoning:*

1. Happy hardcore is a music genre that is energetic and upbeat. $[s_1 = 0.000]$

2. Funerals are traditionally somber events. $[s_2 = 0.000]$

3. The genre is not typically associated with funerals. $[s_3 = 0.000]$

4. Therefore, it is unusual to play Happy hardcore music at a funeral. $[s_4 = 0.000]$

**Answer:** True

*Mean: 0.000*

**Aligned Model (Qwen3-4B-Instruct-2507, LIFT-aligned)**

*Reasoning:*

1. Happy hardcore is a music genre that is associated with energetic, upbeat, and dance-oriented events such as raves. $[s_1 = 0.051]$

2. Funerals are traditionally somber and not associated with dancing. $[s_2 = 0.047]$

3. Therefore, playing Happy hardcore music at a funeral would be unusual. $[s_3 = 0.049]$

**Answer:** True

*Mean: 0.049*

---

This example illustrates the difference in reasoning behavior between the two models. While both models produce correct answers, their reasoning traces have different causal structures. The base model generates four reasoning steps with faithfulness scores of zero (up to 11 digits of precision)—intervening on any step does not change the model's final answer probability. This suggests that the answer is predetermined, with the reasoning serving as a post-hoc explanation rather than causally determining the conclusion.

In contrast, the LIFT-aligned model produces a more concise three-step trace where each step exhibits measurable causal influence (mean faithfulness 0.049). Beyond the quantitative improvement, the aligned model demonstrates qualitative differences in reasoning structure. Each step explicitly builds on preceding information: the first step establishes the genre's association with dance-oriented events, the second introduces the incompatibility with funeral contexts by noting that funerals are "not associated with dancing," and the third synthesizes these facts into the final conclusion. This sequential dependency structure reflects the causal necessity that LIFT optimizes for during training.

## 6. Limitations

While LIFT demonstrates consistent faithfulness improvements, several limitations warrant discussion.

**Computational Cost.** LIFT requires generating multiple modified traces per training example (one per reasoning step), making data generation significantly more expensive than standard supervised fine-tuning. For a trace with $n$ steps, we require $n$ forward passes for scoring plus $n$ generation passes for continuations.

**Base Model Capability Requirement.** LIFT assumes the base model can already generate coherent chain-of-thought reasoning. For models that produce incoherent or very short traces, the intervention-based scoring may not provide meaningful signal. Future work could explore bootstrapping approaches for models with weaker initial reasoning capabilities.

**Fact Corpus Dependency.** Our intervention quality depends on the fact corpus $\mathcal{F}$ containing domain-appropriate facts. For highly specialized domains (e.g. advanced mathematics, medical reasoning), the corpus may lack suitable counterfactuals, potentially reducing intervention effectiveness. Future work could explore alternative methods of retrieving facts for interventions.

**Hyperparameter Selection.** Due to computational constraints, hyperparameters were selected through manual tuning on a small development set rather than systematic grid search. The relationships between hyperparameters (temperature settings, score thresholds, LoRA rank) and final performance remain underexplored. Future work could investigate these dependencies more rigorously to optimize the faithfulness-accuracy trade-off and identify which hyperparameters are most critical for different model families or task types.

## 7. Future Work

Several promising directions extend LIFT's capabilities.

**Multi-Step Interventions.** Current scoring intervenes on single steps independently. Future work could explore joint interventions on multiple steps simultaneously, measuring whether step combinations are collectively necessary. This could reveal redundancy in reasoning traces and identify minimal sufficient subsets.

**Adaptive Intervention Strategies.** Rather than using a fixed fact corpus, future methods could generate counterfactual edits on-the-fly using language models, conditioning on the specific reasoning context. This could improve intervention quality in specialized domains and adapt to novel reasoning patterns.

**Scaling to Multi-Modal Reasoning.** As language models increasingly incorporate vision and other modalities, extending LIFT's intervention-based approach to multi-modal reasoning traces presents an important challenge. This would require developing appropriate counterfactual interventions for visual evidence and other non-textual reasoning components.

## 8. Software and Data

For reproducibility we provide the complete list of hyperparameters, prompt templates, the fact corpus and cluster assignments, and data files containing the examples used for fine-tuning the models. We provide code to generate examples using LIFT and fine-tune for faithfulness, or fine-tune directly on the examples that we used. All files can be found in the supplementary material.

## Impact Statement

This paper presents work whose goal is to advance the field of Machine Learning. There are many potential societal consequences of our work, none which we feel must be specifically highlighted here.

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

## A. Fact Corpus Construction and Clustering

**Corpus Sources.** Our fact corpus $\mathcal{F}$ contains approximately 350,000 entries from two sources:

- **Wikidata facts** (250,000 entries): We extract factual triples from Wikidata and convert them to natural language using a language model. Examples include geographic facts ("Tokyo is the capital of Japan"), temporal facts ("World War II ended in 1945"), and relational facts ("Albert Einstein developed the theory of relativity").

- **Arithmetic equations** (100,000 entries): We generate simple arithmetic expressions covering addition, subtraction, multiplication, and division with operands ranging from $-1000$ to 1000. Each equation is formatted in multiple styles: symbolic ("143 - 61 = 82"), natural language ("105 + 106 equals 211."), and mixed formats.

**Embedding and Clustering.** We compute sentence embeddings using the BGE-large-en-v1.5 model, which produces 1024-dimensional vectors. We apply k-means clustering with 20 initial clusters and $K = 5000$ clusters, chosen to balance granularity (enabling domain-specific matching) and coverage (ensuring sufficient facts per cluster).

The same fact corpus was used for interventions for all models and runs. All facts and clusters are provided in the codebase.

## B. Style Rewriting Implementation

To maintain coherence with the original CoT context, we rewrite the retrieved counterfactual fact $f^*$ to match the writing style of the original step $r_i$. We use the base model itself to perform this rewriting via few-shot prompting.

**Prompt Structure.** The prompt contains 16 few-shot examples demonstrating various style transformations:

- Formal to informal: "Seven plus two equals nine" $\rightarrow$ "In math class today, we discovered that seven plus two makes nine"

- Symbolic to code: "6 × 7 = 42" $\rightarrow$ "# compute product\nresult = 6 * 7"

- Natural language to symbolic: "Nine minus four equals five" $\rightarrow$ "9 - 4 = 5"

- LaTeX formatting: "8 / 2 = 4" $\rightarrow$ "$8 \div 2 = 4$"

The prompt explicitly instructs the model to rewrite the step such that its writing style flows with the original step but the content contradicts it. This ensures $\tilde{r}_i$ is contextually coherent but factually incorrect.

**Generation Parameters.** We use greedy decoding for deterministic style rewriting. The model generates a chain-of-thought explanation, followed by the final rewritten step on a separate line, which we extract via pattern matching.

## C. Static Faithfulness Metric

In addition to our main faithfulness metric (which regenerates continuations), we evaluate *static faithfulness*: the log-probability drop when replacing a step but keeping all subsequent steps unchanged.

**Definition.** For step $i$ in trace $\mathbf{r} = (r_1, \ldots, r_n)$, static faithfulness is:

$$s_i^{\text{static}} = \log p(a \mid q, \mathbf{r}) - \log p(a \mid q, r_1, \ldots, r_{i-1}, \tilde{r}_i, r_{i+1}, \ldots, r_n) \tag{8}$$

where $\tilde{r}_i$ is the intervened step from Algorithm 1.

**Interpretation.** Static faithfulness measures whether a step's specific content affects the answer, even when forcing the same continuation. High static faithfulness indicates the model uses the step's information to support the answer. Low static faithfulness suggests the step is decorative—changing its content doesn't affect answer probability because subsequent steps override it.

**Comparison.** Our main (regenerative) faithfulness metric typically yields higher scores because it allows the model to naturally diverge after intervention. Static faithfulness is more conservative, only measuring direct influence on the answer. In particular, if the model's answer is calculated in the final step of the CoT, there is little to no drop in log probability of the answer if a previous step is modified. Nevertheless, we report both metrics in our evaluation to provide complementary views of causal importance.

# D. Reproducibility Notes

Data generation is parallelized across multiple worker threads to improve efficiency. For Mistral-7B-Instruct-v0.3, we use two worker threads on an NVIDIA H100 SXM GPU (80GB VRAM). For Qwen3-8B, we use two worker threads on an NVIDIA GH200 GPU (96GB VRAM). For Qwen3-4B-Instruct-2507, we use three worker threads on an NVIDIA GH200 GPU.

Data generation takes approximately 24 hours (wall-clock time) for Mistral-7B-Instruct-v0.3 and Qwen3-8B, and approximately 10 hours for Qwen3-4B-Instruct-2507. Fine-tuning on the generated 1000 examples takes approximately 10 minutes for Mistral-7B-Instruct-v0.3 and Qwen3-8B, and approximately 6 minutes for Qwen3-4B-Instruct-2507. Total end-to-end training time (including data generation and fine-tuning) is approximately 24 hours for the 7-8B models and 10 hours for the 4B model on the specified hardware.

Evaluation of 300-480 examples per dataset with faithfulness scoring takes approximately 4-5 hours per model per dataset. The total computational cost for reproducing all experiments (3 models, 3 datasets) is approximately 100 GPU-hours on H100/GH200 class hardware.