# OpenReview forum: "LIFT: Likelihood-Based Intervention Faithfulness Training for Chain-of-Thought in Language Models"
_ICML.cc/2026/Conference — Submitted to ICML 2026_

### Official Review · Reviewer_91C3 · 2026-03-03

**Soundness:** 2
**Presentation:** 3
**Significance:** 2
**Originality:** 2
**Overall Recommendation:** 2
**Confidence:** 3

**Summary:**

This paper proposes LIFT, a fully automated training procedure to improve chain-of-thought (CoT) faithfulness. The key idea is to estimate the importance of individual reasoning steps using counterfactual interventions: replace a given step with a domain-matched but purportedly irrelevant/contradictory alternative and measure the resulting drop in the log-probability of the original final answer. These step scores are then used to select and weight intervened traces for fine-tuning under a weighted objective with KL regularization, with a hyperparameter controlling the faithfulness–accuracy trade-off. Experiments on three LLMs (Mistral-7B-Instruct, Qwen3-8B, Qwen3-4B) across GSM8K, SVAMP, and StrategyQA report improved regenerative faithfulness and static faithfulness, while claiming broadly similar accuracy to CoT baselines.

**Compliance With Llm Reviewing Policy:**

Affirmed.

**Final Justification:**

As no further rebuttal has been provided, I choose to maintain my score.

**Key Questions For Authors:**

1. What proportion of interventions are explicitly contradictory vs merely irrelevant vs incoherent? Please provide a labeled audit.
2. Does LIFT change CoT length or step count distribution? If so, do gains persist under length-controlled comparisons?
3. Have you evaluated faithfulness using a different intervention family to test generalization?
4. What exact statistical test supports the significance claims, and is it paired per-example?
5. How stable are results across multiple fine-tuning seeds?

**Limitations:**

- While some practical limitations are acknowledged, the paper does not adequately discuss:
- Coupling between training and evaluation interventions,
- Lack of intervention-quality validation,
- Step-length confounds in the metric,
- The risk of over-interpreting perturbation sensitivity as causal dependence,
- Presentation-level concerns affecting reproducibility clarity.

**Strengths And Weaknesses:**

**Strengths**
- Coherent end-to-end pipeline: intervention generation → step scoring via log-prob drops → weighted fine-tuning objective with KL regularization.
- Experiments span multiple models and datasets, reporting both regenerative and static faithfulness.
- Ablation on answer-weight scaling illustrates a controllable faithfulness–accuracy trade-off.

**Weaknesses**
- The “causal influence” framing is not rigorously justified. The score is a model-based perturbation sensitivity metric under a specific intervention and decoding regime, and may conflate semantic corruption, distribution shift, and sampling noise.
- Evaluation is tightly coupled to the training mechanism: the same intervention generator is used for both training and evaluation, raising concerns of circularity.
- Intervention quality is not validated. There is no quantitative audit of whether interventions are irrelevant, contradictory, or nonsensical, nor how often they preserve correctness.
- Step-count and verbosity confounds are not controlled. Since faithfulness is averaged over steps, metric shifts may reflect segmentation or length changes rather than genuine semantic improvements.
- Statistical testing is underspecified despite “statistically significant” claims.
- Small-data fine-tuning (1000 prompts, 1 epoch, hand-tuned hyperparameters) lacks multi-seed stability analysis.

---

### Official Review · Reviewer_CxFT · 2026-03-12

**Soundness:** 2
**Presentation:** 3
**Significance:** 2
**Originality:** 2
**Overall Recommendation:** 2
**Confidence:** 4

**Summary:**

The paper introduces LIFT (Likelihood-based Intervention Faithfulness Training) to address unfaithfulness in LLM Chain-of-Thought reasoning, where models generate reasoning steps that do not actually drive the final answer. The method utilizes an automated counterfactual intervention process, replacing reasoning steps with domain-matched but irrelevant facts to measure the subsequent drop in the answer's log-probability. This drop serves as a causal faithfulness score, identifying which steps are truly necessary for the model's conclusion. These scores then guide a LoRA-based fine-tuning objective that jointly optimizes for faithfulness and accuracy via a tunable hyperparameter, $\eta$. By weighting training samples according to their causal influence, LIFT forces the model to rely more heavily on its stated reasoning. Evaluation across Mistral and Qwen models on benchmarks like GSM8K and StrategyQA shows relative faithfulness gains of 20% to 250% without compromising task performance.

**Compliance With Llm Reviewing Policy:**

Affirmed.

**Key Questions For Authors:**

1. Baseline Comparisons: Why were no existing automated methods (like FRODO or basic CoT filtering based on outcome) included in the Results table (Table 2)? A quantitative comparison would better situating LIFT's performance.

2. Algorithm 2 intervenes on one step at a time. Do the authors have data on whether LIFT-aligned models show improved faithfulness for interactions between steps, or does it primarily optimize for the local dependency of the answer on each individual step?

3.  Table 1 shows widely varying $\eta$ values (0.4 to 3.0). Is there a heuristic for choosing this value without a labeled validation set for faithfulness, which the paper claims to avoid needing?

4. Can the authors discuss  some recent research and add more competitive baselines?

**Limitations:**

The paper’s theoretical foundation is somewhat undermined by a thin discussion of the underlying mechanisms that cause unfaithfulness in the first place. While the authors correctly identify that reasoning steps are often "decoupled" from the final answer, they miss an opportunity to engage with established literature explaining why this occurs—such as the tendency for LLMs to rely on internal priors or "shortcuts" rather than the provided context. Without a deeper dive into the architectural or training biases that lead to post-hoc rationalization, the motivation for the specific design of LIFT remains primarily empirical rather than grounded in a comprehensive diagnostic framework of model behavior.

Furthermore, the submission suffers from a significant lack of contemporary context, as the majority of the cited literature is nearly two years old. In the rapidly evolving landscape of LLM interpretability and reasoning, the omission of recent 2024 and 2025 research on self-correction, process-based reward modeling, and advanced preference optimization is a notable oversight. This reliance on older references makes it difficult to assess how LIFT compares to the current state-of-the-art in automated faithfulness improvement and leaves the work feeling disconnected from the most recent advancements in the field.

**Strengths And Weaknesses:**

Strengths:

1. The automated use of "domain-matched but irrelevant" facts for counterfactual intervention is a clever way to bypass the need for expensive human-labeled rationales or "gold" faithful traces.

2. The introduction of the $\eta$ parameter (and the $\lambda$ ablation) provides a clear mechanism for developers to balance the tension between strict causal faithfulness and raw task performance.

3. The inclusion of "Static Faithfulness" as a secondary, more conservative metric (keeping subsequent steps fixed) adds significant depth to the evaluation and confirms that the model is truly learning step-dependency rather than just becoming more sensitive to noise.


Weeknesses:

1. As noted in the limitations, the scoring phase requires $n$ forward passes and $n$ generation passes for a trace of $n$ steps. This significantly increases the "cost per training token" compared to standard SFT.

2. The paper compares "Aligned" vs. "Base" models. While it mentions FRODO and PRM800K in the introduction, it lacks a direct head-to-head empirical comparison with other automated faithfulness-improving methods.

3. The paper’s theoretical foundation is somewhat undermined by a thin discussion of the underlying mechanisms that cause unfaithfulness in the first place. While the authors correctly identify that reasoning steps are often "decoupled" from the final answer, they miss a critical opportunity to engage with established literature explaining why this occurs.

4. The paper lacks a discussion of contemporary research within the Related Work section, with the majority of cited literature being more than two years old. This is a significant oversight given the rapid evolution of the field. For instance, recent studies  in 2025 have specifically highlighted how autoregressive training objectives can induce "confirmation bias" and "implicit post-hoc rationalization", but there is no mention in the paper.

---

### Official Review · Reviewer_UVKb · 2026-03-12

**Soundness:** 2
**Presentation:** 2
**Significance:** 2
**Originality:** 2
**Overall Recommendation:** 2
**Confidence:** 4

**Summary:**

This paper proposes a finetuning method for improving the CoT faithfulness of LLMs. They first propose a metric for measuring faithfulness of each step of a CoT by introducing synthetic interventions, and then they use this metric as a way to finetuning a model. They show that their approach increases faithfulness with minimal impact on answer accuracy.

**Compliance With Llm Reviewing Policy:**

Affirmed.

**Final Justification:**

There was no rebuttal, so my original assessment holds.

**Key Questions For Authors:**

1. What are the limitations of the faithfulness metric? I think the metric basically encourages pruning away unnecessary information for instance if (A or B => C), then the CoT may only include fact A since after A is included, adding B may not increase the probability of C being predicted. Why is this the correct thing to do? The answer to this question will help me understand the justification for the metric and method proposed in the paper.
2. Why does accuracy decrease with decreasing lambda? Does this point to a limitation of the finetuning approach (as in higher faithfulness is not better) or is there some other reason why this happens?

**Limitations:**

yes

**Strengths And Weaknesses:**

### Strengths:
**Soundness:**
* Results are evaluated on three models and three datasets showing significant improvements in faithfulness while sometimes even increasing accuracy.
* Errors are reported in Table 2.
* Limitations are discussed.
* An ablation on one of the hyperparameters is performed showing a tradeoff between faithfulness and accuracy.

**Presentation:**
* The work is generally well structured and could probably be reproduced since further details are provided in the appendix.

**Significance:**
* The problem of improving CoT faithfulness is very important to the trustworthiness of LLMs.

**Originality:**
* The approach taken in this paper of directly optimizing a faithfulness metric through generating synthetic interventions on CoTs appears novel and is well motivated.

### Weaknesses:
**Soundness:**
* It is possible that a CoT includes irrelevant information as can happen if the model tries solving the problem one way and then backtracks to solving it a separate way. In this case, modifying the facts in the first solution attempt may have no impact on answer likelihood which could be a limitation of this measure of faithfulness. This is briefly mentioned in relation to s_low and s_high, but how are these thresholds chosen?
* I think there is a typo in equation 4 where s_i^j should be s_j^i. This appears to be carried forward into line 5 of Algorithm 3.
* While this approach shows a tradeoff between faithfulness and accuracy, it is not obvious why that should be the case and leaves me wondering if this is a general phenomenon or just a limitation of the approach taken here. I would think that intuitively a more faithful model should achieve higher accuracy, but if this isn’t the case, then what if you train to actually decrease faithfulness, then can you increase accuracy?

**Presentation:**
* Several of the citations are missing dates or other information (Barez et al, Hu et al).
* While a related work section is not necessary, more discussion of the positioning with related work is needed. Currently, for the CoT faithfulness problem there are only two citations and these miss some of the early work such as [1], and [2].

[1]: Language Models Don't Always Say What They Think: Unfaithful Explanations in Chain-of-Thought Prompting. Miles Turpin, Julian Michael, Ethan Perez, Samuel R. Bowman. NeurIPS 2023.

[2]: Faithful Chain-of-Thought Reasoning. Qing Lyu, Shreya Havaldar, Adam Stein, Li Zhang, Delip Rao, Eric Wong, Marianna Apidianaki, Chris Callison-Burch. AACL 2023.

**Significance:**
* The original release of o1 noted that they chose to hide the CoT outputs and wanted to avoid explicitly training on the CoT or making the CoT conform to certain policies [1]. Now most frontier models have their CoT hidden so it is unclear how this approach would apply to reasoning models (which are commonly used to answer reasoning problems).

[1]: https://openai.com/index/learning-to-reason-with-llms/

**Originality:**
* The proposed faithfulness measure is not compared to existing approaches to measuring faithfulness. It would be nice to see if it at least aligns or what the main differences are.

---

### Official Review · Reviewer_BA5A · 2026-03-13

**Soundness:** 1
**Presentation:** 3
**Significance:** 1
**Originality:** 3
**Overall Recommendation:** 2
**Confidence:** 4

**Summary:**

This paper proposes a method for training LLMs to have more 'faithful' chains-of-thought, meaning they actually use their reasoning steps in producing the final answer (making CoTs 'causally necessary'). The method works by fine-tuning on CoT traces where reasoning steps are measured to be causally important for the final answer, jointly training for accuracy at the same time. To measure the causal importance of reasoning steps, the method requires access to a corpus of facts relevant for the particular domain trained on. The authors find improved faithfulness for 3 small LLMs (Mistral-7B, Qwen3-8B, and Qwen3-4B) in 3 domains (GSM8k, SVAMP, StrategyQA) without sacrificing accuracy.

**Compliance With Llm Reviewing Policy:**

Affirmed.

**Final Justification:**

I remain of the opinion this paper should be rejected because the experiments are not sound. The authors did not provide a rebuttal so my opinion did not change after the discussion period.

**Key Questions For Authors:**

- Can you compare to the base model trained for accuracy on the same questions without faithfulness?
- Can you demonstrate generalisation to other measures of faithfulness?
- Can you demonstrate how much the method relies on the fact corpus?

**Limitations:**

Yes

**Strengths And Weaknesses:**

**Strengths**

**Presentation**: The paper is clearly written and easy to follow. The experimental setup and results are presented with sufficient detail to understand the work.

**Significance**: Faithfulness of reasoning traces is a core challenge for interpretability and oversight of LLM reasoning, and the problem framing here is sharp and well-motivated.

**Originality**: The proposed method is both novel and parsimonious, making it a natural fit for the evaluation task.

**Weaknesses**.

*Soundness*:
- The main finding that faithfulness can be increased without harming accuracy is not exactly sound in my opinion. The authors show that if you train for faithfulness without jointly optimising for accuracy, the accuracy drops. Indeed, when jointly optimising for accuracy and faithfulness the accuracy stays the same, but the relevant baseline to compare this to is not the base model but the base model finetuned for accuracy alone.
- The measure of causal importance of the reasoning step (faithfulness) is relative to a reasoning trace with a counterfactual fact planted into the reasoning trace. While this does demonstrate increased likelihood of the answer due to putting back the original reasoning step, there's a confounding factor of distribution shift due to the implanted counterfactual. Since all scores have this same issue, maybe it does not matter, but at the same time the score can be interpreted as a kind of 'robustness to implanted counterfactuals', which is different from faithfulness. Intuitively, maybe the original reasoning step was not causally important for the answer, but the implanted counterfactual itself was so harmful that the likelihood of the answer nonetheless dropped. The qualitative example in 5.2 also makes me suspect this is the case, because in both the original trace and the aligned trace the model is essentially saying the same things but differently presented.

*Significance*:
The reliance on the fact corpus as well as the fact that you need access to reasoning examples with labels to not deteriorate in accuracy makes this method less likely to be applicable especially in areas where faithfulness is important (e.g. when we need the reasoning trace to understand what the model is doing, if we do not know the accuracy or validity of the answer but are trying to judge it based on the reasoning trace).
The method uses the faithfulness metric used for training as evaluation of improved faithfulness as well. It would be more significant if LIFT would generalise to other ways of measuring faithfulness (e.g. the ability to predict the answer from the CoT).
Finally, the actual increase in faithfulness from LIFT training is limited, likely because of the trade-off with accuracy.

Nits:
- missing year in citation Barez et al in the intro.
- You mention that most methods improving faithfulness only work on small models in the abstract, but your method is also only tested on small models.
- Figure 3 is quite difficult to read, would consider changing the presentation of this plot.

---

### Decision · Program_Chairs · 2026-04-30

**Decision:**

Reject

**Comment:**

This paper introduces a fine-tuning method to improve the faithfulness of chain-of-thought reasoning via a likelihood-based intervention metric. The problem is important and the paper is generally clear. However, all reviewers raise substantial concerns that limit the strength of the submission.

A key issue is the validity of the faithfulness metric, which may conflate causal dependence with robustness to perturbations, making improvements hard to interpret. The claim of maintaining accuracy is also weak due to missing baselines and an unclear trade-off. Evaluation is limited to small models and the same metric for training and testing, with little evidence of generalization. The method also requires domain-specific data and high compute, raising concerns about practicality.

Overall, while the idea is promising, the paper does not provide sufficiently convincing evidence to support its claims.